

# Gaussian wake model fitting in a transient event over Alpha Ventus wind farm

Maria Krutova[1,2] and Mostafa Bakhoday-Paskyabi[1,2]

[1]Geophysical institut, University of Bergen, Allégaten 70, 5007 Bergen, Norway
[2]Bergen Offshore Wind Centre, University of Bergen, Allégaten 55, 5007 Bergen, Norway

**Correspondence:** Maria Krutova (maria.krutova@uib.no)

**Abstract.**

Engineering wake models are defined by mathematical expressions and a set of coefficients. Because of their simplicity, the models may be rigid in transient events such as open cellular convection (OCC), characterized by a strong wind speed and direction change within tens of minutes. We use the results of a multiscale wind-wake modeling during an OCC event at the
Alpha Ventus wind farm in the Southern North Sea to study how Gaussian models capture wake deficit variabilities. We find that the Jensen-Gaussian model would benefit from a constant coefficient tuning. On the contrary, the Bastankhah and Porté-Agel model and the super-Gaussian model are consistent without tuning but perform best with different deficit distribution shapes.

## 1 Introduction

Engineering wake models predict wake deficit with simple analytical expressions and known values of the free-flow characeristics and thrust coefficient. Models' simplicity ensures fast calculation but smooths instantaneous wake features. The wind speed, turbulence intensity and thrust coefficient provide the model with limited capability to adapt to the flow. Since strong changes in the flow may affect the accuracy of the prediction, the model's reaction to transient events has to be studied (Bakhoday-Paskyabi et al., 2022a, b).

Transient events change the flow characteristics within several minutes, which makes numerical simulations unfeasible for real-time control. Although engineering models do not resolve the instantaneous wake structure, they can operate on a steady averaged flow before and after the event.

Open cellular convection (OCC) is a transient event associated with a cold-air outbreak above the warm ocean surface. We regard an OCC event in the Southern North Sea on November 22, 2015, near FINO1 platform (Fig. 1, Sect. 2). We select
three Gaussian models (Sect. 3) and evaluate their performance before and after the OCC event based on the accuracy of predicting the 10-minute average normalized wake deficit in the downstream cross-sections (Sect. 4). We also evaluate whether the prediction can be improved if the model's coefficients are corrected by choosing the best fit for the wake deficit.

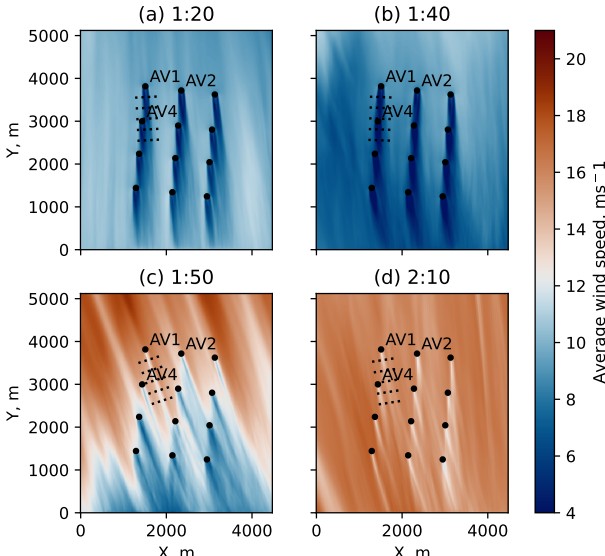

**Figure 1.** The cross-section of the innermost domain of the WRF-LES containing the Alpha Ventus wind farm. The cross-section shows 10-minute averages of the wind speed at the hub height; time stamps mark the end of each 10-minute period. a) Flow before the OCC event; b) the OCC event is about to begin, the wind farm is not yet affected; c) the flow in the wind farm undergoes radical changes due to the OCC event; d) the convective cell had passed the wind farm, the flow is stabilizing.

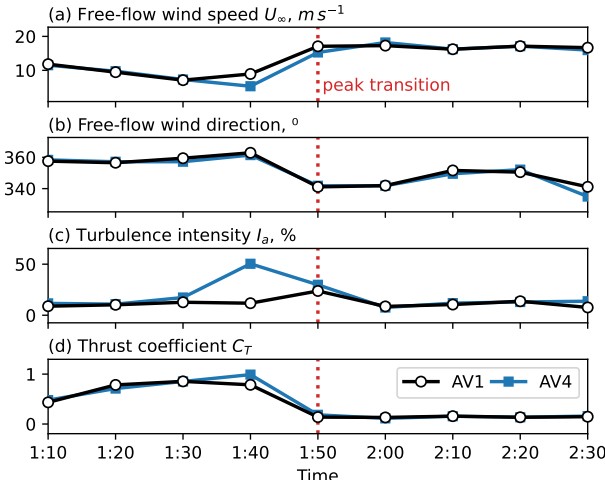

**Figure 2.** 10-minute averaged inflow characteristics at the hub height $2D$ upstream of AV1 and AV4 (a-c) and the thrust coefficient (d).



## 2 Data

The convective cell passed the FINO1 platform and Alpha Ventus wind farm on November 22, 2015 $1:40-2:00$ AM UTC+0

(Fig. 1). Due to the lack of lidar data for this day, we utilize a high-fidelity numerical simulation to reproduce the flow field (Bakhoday-Paskyabi et al., 2022b). The Weather Research and Forecasting (WRF) model output acts as a dynamic driver input for the large-eddy simulation (LES) PALM model system (Maronga et al., 2020). The LES consists of two nested domains, with the inner domain $4480 \times 5120 \times 320\,\mathrm{m}$ refining the grid around Alpha Ventus to the grid cell size of $\Delta = 10 \times 10 \times 5\,\mathrm{m}$.

The passing open cell affects the flow characteristics changing the wake shapes. We define three phases. The pre-OCC phase covers time stamps $1:10-1:40$ and is characterized by low wind speed but high thrust (Fig. 1ab); the wakes are wide and merge for turbines of the same column. A short transition phase takes about 10-minutes and is most prominent at 1:50 time stamp (Fig. 1c). The post-OCC phase starts at 2:00 time stamp and is characterized by low thrust, but high wind speed (Fig. 1d); the wakes become narrow and do not merge anymore. The turbulence intensity fluctuates withing $7-11\%$ for both phases, the

wind direction changes by $10-15°$ between phases.

The Alpha Ventus wind farm consists of 12 wind turbines of two types arranged in a rectangular pattern, with the turbine AV1 located in the northwest corner. All turbines are approximated as the NREL 5MW reference turbine (Jonkman et al., 2009) and actuator disc model with rotation (Witha et al., 2014); the turbines are set to the same hub height $z_h$ of 90 m and rotor diameter $D$ of 126 m.

## 3 Methodology


### 3.1 Model fitting

Model fitting is performed by taking WRF-LES results as a true value. We choose the turbine AV1 as a reference for the regarded wake models since AV1 is the least affected by the nearby wakes due to the northwest wind. To extract the cross-sectional velocity $U$ in $xy$-plane, we define a local coordinate system with a center at the AV1 position so that the $x$-axis always

follows a 10-minute average wind direction. The downstream cross-sections are regarded in a range of $x/D = 2..10$ with a step of $0.5$ – totaling 17 cross-sections. The local coordinates for each cross-section are represented by a radial distance $r$ varying in the range of $r/D = -2..2$ from the rotor axis. The 10-minute average inflow characteristics – wind speed $\overline{U}_0$, wind direction, and turbulence intensity $I_a$ – are estimated by probing the free flow at $2D$ upstream of AV1. The PALM LES output contains the thrust force $T$ for each turbine; hence, the thrust coefficient $C_T$ is derived as

$$C_T = \frac{T}{0.5 \cdot \rho A \overline{U}_0^2} \tag{1}$$

where $\rho = 1.17\,\mathrm{kg/m^3}$ is the constant air density as returned by the WRF-LES, $A = \pi(D/2)^2$ is the rotor area.

The wake from AV4 appears in AV1 cross-sections starting from $x/D = 6.5$. The inflow probes for AV4 generally return similar values, except for the time stamp 1:40 (Fig. 2) – the wind speed and turbulence intensity near AV4 are strongly affected





**Table 1.** Gaussian wake models and their original coefficients.

| Model | Coefficients |
| --- | --- |
| BPA | $k_1^* = 0.003678$, $k_2^* = 0.3837$ |
| Jensen Gaussian | $k = 0.05$ |
| Super-Gaussian | $a_s = 0.17$, $b_s = 0.005$, $c_s = 0.2$ |

by the direct hit from AV1 wake. Overall, this similarity allows using AV1 inflow characteristics and thrust coefficient to
estimate wakes for both turbines. An ensemble wake is calculated by summing up the normalized wake deficit from the AV1
wake at the regarded cross-section and the deficit at the respective cross-section of the AV4 wake.

We also tune coefficients of each model to find the best fit to WRF-LES results. Overall, we regard the following cases:

- **default model** – a wake model uses coefficients suggested by its authors and relies only on $C_T$ and $I_a$ to calculate wake
  deficit.

- **corrected fit** – model's coefficients are fitted for the first 10-minute period of the pre-OCC and post-OCC phases –
  periods ending at 1:10 and 2:00, respectively – and remain fixed for the further periods of the phase.

- **best fit** – model's coefficients are re-fitted to each new 10-minute period. The fit uses already known simulation data for
  the passing period, so while not being practical, it shows whether the model could have described the wake better.

The best fit is optimized for all cross-sections in a 10-minute period to avoid tuning models to a specific part of the wake. To
evaluate the models' performance, we also compare root mean square errors (RMSE) for the normalized wake deficit of each
cross-section separately.

### 3.2 Gaussian wake models

We apply engineering wake models to calculate normalized wake deficit $\Delta \overline{U} = 1 - U/U_0$. Since we are interested in how well
wake models approximate the wake shape and flow characteristics, we select wake models that suggest a Gaussian distribution
of the deficit. The three Gaussian models regarded in this study are chosen based on their flexibility and possibility to re-fit the
parameters (Krutova et al., 2020). When choosing which values to fit, we prefer the coefficients fitted by the original authors
(Table 1).

#### 3.2.1 BPA Gaussian model

One of the first Gaussian models was developed by Bastankhah and Porté-Agel, further referred to as the BPA model (Bas-
tankhah and Porté-Agel, 2014). We utilize Niayifar and Porté-Agel (2016) version of the model, which parameterizes the





growth rate of the wake $k^*$ as a linear function of the turbulence intensity $I_a$:

$$k^* = k_1^* + k_2^* I_a, \tag{2}$$

where $k_1^*$ and $k_2^*$ are the coefficients to fit. Then the standard deviation $\sigma$ of the Gaussian function depends on the diameter $D$ and downstream distance $x$ as

$$\sigma = k^* x + \varepsilon D, \tag{3}$$

$$\varepsilon = 0.2\sqrt{\beta}, \ \beta = \frac{1}{2}\frac{1+\sqrt{1-C_T}}{\sqrt{1-C_T}} \tag{4}$$

The normalized wake deficit is then given as

$$\Delta\overline{U} = \left(1 - \sqrt{1 - \frac{C_T}{8(\sigma/D)^2}}\right) \times \exp\left(-\frac{r^2}{2\sigma^2}\right) \tag{5}$$

### 3.2.2 Jensen-Gaussian model

Gao et al. (2016) replaced a top-hat distribution in the Jensen wake model (Jensen, 1983) with a Gaussian.

$$\Delta\overline{U} = \left[1 - \frac{5.16}{\sqrt{2\pi}} \cdot \overline{U}^*(x,k_w)\right] \times \exp\left(-\frac{r^2}{2\sigma^2}\right) \tag{6}$$

$$\sigma = r_x/2.58 \tag{7}$$

where $r_x(C_T, k_w)$ is the wake radius, $\overline{U}^*(x,k_w)$ and $k_w(k,C_T,I_a)$ are new functions omitted here for the sake of brevity. The full definition can be found in the original study (Gao et al., 2016).

The only coefficient to fit, the wake decay coefficient $k$, is defined similarly to the Jensen model and is then corrected with the function $k_w(k,C_T,I_a)$. We choose the starting value $k = 0.05$ as suggested by the model's authors for offshore wind farms.

### 3.2.3 Super-Gaussian model

A regular Gaussian model uses the order of $n = 2$. Increasing this value leads to the tendency for a flat peak of the distribution. Blondel and Cathelain (2020) proposed a super-Gaussian model with varying $n$. Although another version of this model

exists (Cathelain et al., 2020), it mainly improves the wake deficit prediction for $x/D < 5$, which we already found satisfactory for our purposes. Hence, we prefer the first version with less parameterized coefficients.

The super-Gaussian model expands the BPA model and alters the order $n$ depending on the downstream distance $x$

$$\Delta\overline{U} = C(x)\exp\left(-\frac{|r|^{n(x)}}{2\sigma^2}\right) \tag{8}$$

where $C(x)$ is defined via $n(x)$ as

$$C(x) = 2^{2/n-1} - \sqrt{2^{4/n-2} - \frac{nC_T}{16\Gamma(2/n)\sigma^{4/n}}} \tag{9}$$

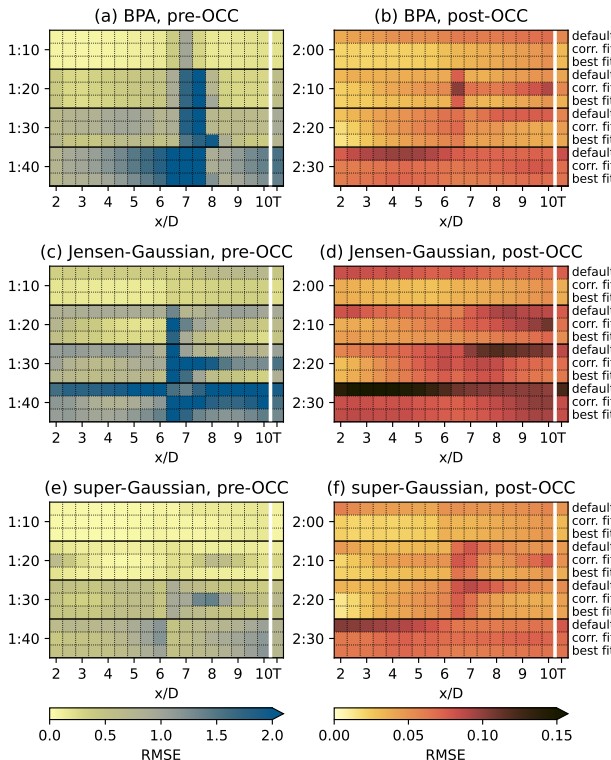

**Figure 3.** RMSE of the Gaussian models calculated for each cross-section. The label 'T' stands for the total RMSE normalized by the number of cross-sections to allow comparison. The results for the peak transition time 1:50 are excluded since the RMSE is not representative there.

and collapses to the multiplier in Eq. (5) for $n = 2$.

The standard deviation $\sigma$ in this model is defined as

$$\sigma = (a_s I_a + b_s) + c_s \sqrt{\beta} \tag{10}$$

Here, $\beta$ follows Eq. (4) of the BPA model with $c_s = 0.2$. $a_s$ and $b_s$ are defined differently despite being used similarly to $k_2^*$

and $k_1^*$ in Eq. (2). We fit these three coefficients (Table 1).

The super-Gaussian model proposes two method of finding $n(x)$: root-solving and analytical. The analytical method adds three more coefficients; therefore, we choose the root-solving method and find $n(x)$ from the equation

$$C(x)^2 - 2^{2/n} C(x) + \frac{n C_T}{16 \Gamma(2/n) \sigma^{4/n}} = 0 \tag{11}$$

## 4 Results

The general behavior of all models follows similar trends (Fig. 3). The agreement to WRF-LES is good for the stabilized flow in the pre-OCC phase but declines as the convective cell approaches the wind farm. During the peak transition phase at 1:50,

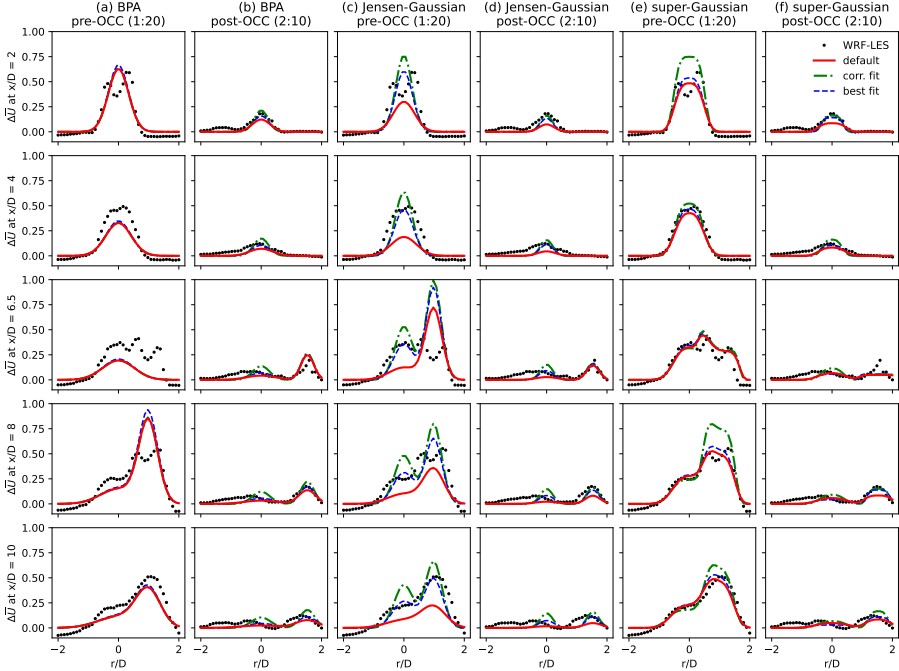

**Figure 4.** Gaussian wake models applied to the normalized wake deficit at the various downstream distances. For $x/D \geq 6.5$, two wakes are present in a cross-section.

the free-flow wind speed becomes ambiguous. The near wake may already be affected by the convective cell – probing the flow upstream would return the correct free-flow wind speed for the near wake. At the same time, the effects of the accelerated flow had not yet reached the far wake (Fig. 1c) – the upstream flow characteristics are not relevant. Probing the free flow at a
distance parallel to the wake is not accurate enough due to wake effects from the nearby turbines. Model fitting for the period ending on 1:50 returns unrealistic coefficients, e.g., a negative wake decay coefficient $k$ for the Jensen-Gaussian model.

The post-OCC phase is less challenging for the models because the wakes no longer merge due to the changed wind direction and narrower wake deficit distribution. However, despite the averaging, the deflection effect becomes more prominent in the far wake. We did not account the deflection, hence, the radial positions of the wake deficit maximums do not perfectly
match maximums extracted in cross-sections for the time stamp 2:30. The discrepancy between maximums positions leads to increased RMSEs, although the agreement in earlier periods was acceptable.

Depending on $C_T$ and $I_a$, the BPA model may not resolve the near wake ($x/D < 3$) due to a negative value occurring under the square root in Eq. (5). While this is not crucial for the AV1 wake, the model may miss the influence from the AV4 wake for several cross-sections leading to an increased RMSE for $x/D \geq 6.5$ (Fig. 3a, 4a). This complicates searching for a best fit – the
coefficients fitted for the BPA model do not deviate much from the original values, and the best fits do not gradually improve RMSEs compared to the default model. Overall, the BPA model tends to overestimate the maximum deficit in the near wake, especially if the wake deficit distribution has a double peak shape.





The Jensen-Gaussian model benefits from correcting the wake decay coefficient $k$ similarly as the Jensen model requires adjustments for better prediction (Peña et al., 2016). The chosen value of $k = 0.05$ underestimates the pre-OCC wake deficit

implying that the initial choice requires re-evaluation based on the observed wake deficit (Fig. 4c). The corrected fit may follow the wake deficit better but tends to a narrow distribution underestimating the wake width and overestimating the maximum wake deficit. Despite dependencies on the thrust coefficient and turbulence intensity, having only one adjustable coefficient limits the flexibility of the Jensen-Gaussian model. In addition, the only coefficient is sensitive to short-term fitting: $k$ varied between $-0.01$ and $0.03$ for the regarded periods, although showing a tendency for $k = 0.01$ in both phases.

Both Jensen-Gaussian and BPA models are subjected to increased RMSE for $x/D \geq 6.5$ in the pre-OCC phase when the wake from AV4 enters the AV1 downstream cross-section. The models either overestimate the deficit in the AV4 wake or underestimate it in the AV1 wake. A notable exception is the super-Gaussian model, which follows the complex shape of the merging pre-OCC wakes well (Fig. 4e).

The tendency for a flat distribution in the super-Gaussian model smooths the double peak in the near wake and resolves a

single peak in the far wake equally well. Consequently, the RMSE of individual cross-sections is rather uniform for the super-Gaussian model (Fig. 3e). Unlike other models, the super-Gaussian model does not benefit from re-fitting the coefficients in the pre-OCC phase – new fits either only slightly deviates from the default model or noticeably overestimates the near wake deficit. On the other hand, a flat peak hinders the super-Gaussian performance in the post-OCC phase (Fig. 4f) – sharp but low distribution peaks are captured by the super-Gaussian model worse than by other models (i.e., BPA and corrected Jensen-

Gaussian). It should be noted that the calibrated super-Gaussian model (Cathelain et al., 2020) slightly increases the maximum wake deficit predicted in the post-OCC near wake, but underestimates it compared to other models.

## 5 Conclusions

We performed a WRF-LES of a transient event and studied how the Gaussian models describe the 10-minute average wake deficit before and after the event. The transient period remained challenging for all models due to the ambiguity in the free-flow

wind speed.

Having only one coefficient, the Jensen-Gaussian model is simple to fit. Moreover, correcting the coefficient based on the flow characteristics and wind farm site conditions is preferable before working with the model. However, adjusting to short averaging periods may not produce a stable coefficient value. While the Jensen-Gaussian consistently showed an improved RMSE with the best fit, this approach is not feasible since all flow characteristics should be known in advance. We do not

recommend using this model for short periods during transient events.

The default definitions of BPA and super-Gaussian models kept a good agreement with the changing flow and did not benefit from the coefficient tuning. While the BPA model could occasionally return better RMSE with the re-fitted coefficients, the default values performed more consistently. Due to how these models interpret the distribution peak, they work better with different wake shapes. The BPA model approximates well a single sharp peak and may result in an increased RMSE when

two wakes are present in a cross-section. The super-Gaussian model smooths the distribution peak, which appears to be a





good approximation of a double peak or merging wakes. Still, the model underperformed in the post-OCC phase with low and uniform wake deficit.

The super-Gaussian model's capability to capture wake merging accurately is promising for applications where the wake deficit distribution is important. Considering that the model also reacts well to changing conditions, this calls for further model
validation in complex wake-wake interaction cases.

*Code and data availability.* The simulation data and the Python code to reproduce the figures are available at https://doi.org/10.5281/zenodo.8135542 (Krutova, 2023).

*Author contributions.* MK wrote the code for model fitting and processed the results, MBP selected the OCC event to design the WRF-LES setup and provided valuable discussion on the findings.

*Competing interests.* The authors declare that they have no conflict of interest.

*Acknowledgements.* The WRF-LES for this study have been performed as a part of the HIghly advanced Probabilistic design and Enhanced Reliability methods for the high-value, cost-efficient offshore WIND (HIPERWIND) project, which has received funding from the European Union's Horizon 2020 Research and Innovation Programme under Grant Agreement No. 101006689. Resources for simulations were provided by UNINETT Sigma2 – the National Infrastructure for High Performance Computing and Data Storage in Norway (project number
NS9696K)



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
