# Peer review of "Gaussian wake model fitting in a transient event over Alpha Ventus wind farm"

_Wind Energy Science, 2023_

## Referee Comment (RC3)

**Review of WES-2023-79**

*Gaussian wake model fitting in a transient event over Alpha Ventus
wind farm*
Maria Krutova and Mostafa Bakhoday-Paskyabi

**Overview:**

The work described in "*Gaussian wake model fitting in a transient event over Alpha Ventus
wind farm*" endeavors to address one of the major shortcomings of analytical wake models—that they
are built on the assumption of stationarity and cannot be expected to reflect changes in forcing
conditions or in turbine operation. However well understood that assumption is, this does lead to
misleading predictions, or high uncertainty, from many engineering models when compared to
operational data. That said, this work does not attempt to update the underlying assumptions used to
build the wake models. Instead, the authors simply highlight several details about the discrepancies
between some wake models and high-fidelity simulations.

The work in this manuscript is interesting but does not by itself merit publication. If the authors wish to
describe model errors with any statistical certainty, they would need to review a large sample of
dynamic conditions (open-cell convection, and weather fronts, morning/evening transitions), and
ensure that they sample wakes from turbines at different points in their operating curves. Otherwise,
the most that can be said about the wake models is that, "they did not match the wakes predicted by
LES in this case." Expanding the sample size would also help indicate whether there may be
straightforward means of improving the models to accomodate these sorts of dynamic events. Such
model improvements would be enormously valued in the wind energy industry and would be much
more valuable for wind plant design, optimization, and prediction.

**Major Comments:**

The wakes predicted by WRF-LES and by Gaussian models are compared for only a single OCC.
How sure can the authors be that the results in the manuscript are a good representation of the
disagreement between models? I would hesitate to draw any real conclusions about the possible
benefits of model parameter tuning or the underlying cause for differences in wake model predictions
from a single sample.

Similarly, the study looks at a small region of the flow in Alpha Ventus, containing two wakes that
interact strongly. The study design does not take into account that there are several possible
superposition methods that could be used for analytical wake models, or the possibility of added
prediction uncertainty introduced by superposition (see ref 1, below) . Other engineering models exist
that do not rely on a superposition scheme. Have the authors considered using those to limit
uncertainty to only the definition of the wake itself? (see refs 2—4).

*Figure 1* . The cross-section of the inner

What do the small makers downstream of AV1 indicate? Sometimes they overlap AV4 and other times they do not. Do these point out measurement areas? This does not match the 17 cross sections from the simulations described in the text.

*Figure 2*. 10-minute averaged inflow chara

I would be interesting to include AV2 in the subplots of Figure 2. AV2 is already called out in Figure 1 and comparing it to AV1 may highlight the spatial variability in turbine performance as well as inflow characteristics.

*khoday-Paskyabi et al., 2022b). The Weather Research and Forecasting (WRF) model output acts as a dynamicdriver input for the large-eddy simulation (LES) PALM model system (Maronga et al., 2020). The LES consists of twonested d*

Is PALM a normal LES framework to use within the WRF model? It would benefit the readers to explain a bit more of the details in the simulation scheme. For example, how is turbulence information exchanged between the WRF simulation and PALM? Do you use the cell-perturbation method? Have you tested sensitivity? Is the WRF simulation driven with reanalysis data? If so, what is the source?

*Figure 3. RMSE of the Gaussian models cal*

This is an interesting figure and appears to be the main result of the study. However, trying to compare results from the various models and model tunings based on the colors of each cell in the figures makes the actual data feel somewhat arbitrary and qualitative. Perhaps the figure could be replaced with trend lines comparing either the different models or the different tuning strategies. I think this would also provide a better opportunity to discuss the surprising result of the peaks in RMSE for the different models.

**Minor Comments:**

*per-Gaussian model proposes two method of finding n(x): root-solving a*

should be plural, "methods"

*= 0.2√β, β = 121 + √1 −CT√1 −CT (4)

Combine fractions to make equation (4) more readable.

*as= 0.17, bs= 0.005, cs= 0.2by the direct hit from AV1 wake . Overall, this similarity allow*

Rephrase.

**References**

1. Hamilton, Nicholas, et al. "Comparison of modular analytical wake models to the Lillgrund wind plant." *Journal of Renewable and Sustainable Energy* 12.5 (2020).
2. Martínez-Tossas, Luis A., et al. "The curled wake model: a three-dimensional and extremely fast steady-state wake solver for wind plant flows." *Wind Energy Science* 6.2 (2021): 555-570.
3. Bastankhah, Majid, et al. "A vortex sheet based analytical model of the curled wake behind yawed wind turbines." *Journal of Fluid Mechanics* 933 (2022): A2.
4. Bay, Christopher J., et al. "Addressing deep array effects and impacts to wake steering with the cumulative-curl wake model." *Wind Energy Science* 8.3 (2023): 401-419.

---

## Author Comment (AC1)

**Response to the comments**

**Maria Krutova**

**December 22, 2023**

Sorry for the delay. Unfortunately, the original response deadline collided with the delivery of my PhD thesis and other urgent work. I am grateful to the editors for extending the deadline so I would have time to cycle through ideas and see how I can provide better insight into the wake models' behavior in the transient event.

I would also like to thank the reviewers for their valuable comments, which not only helped me improve the article directly but also inspired a few ideas I could use when revising.

The last two comments (RC2 and RC3) raised the problem of the insufficient data and analysis presented. The issue is partially caused by the article being submitted initially as a 'Brief communication.' Because of the strict page count limits, the article was severely cut. For the publication, the article has been reclassified as a regular article. Since the page/figure limit no longer applies, it is now possible to expand it. The revised article will contain a more detailed description of the WRF-PALM set up and regard other turbines in the wind farm.

**Response to the reviewer comments 1 (RC1)**

1. **I think that more work is done in the field (like work done by Tuhfe Göçmen et al 2020 J. Phys.: Conf. Ser. 1618 062014 ) that might be relevant for this study and would be worth of mentioning.**

   Thank you for the tip. When searching for the relevant studies, I focused too much on approximating wakes with analytical models, so I overlooked a few obvious examples.

2. **Bakhoday-Paskyabi et al., 2022b study used high-frequency observations and it would be nice to include a sentence or two on LES performance and reliability. Also, why none of available observations used in that study are not suitable for this study.**

   The available observational data consists of various FINO1 time series which provide only local measurements. The mast data can be certainly used to extract and normalize wake deficit values, but we do not have spatial measurements of the flow field.

3. **What about other wake models? I think it would be nice to include a sentence or two about other available wake modes and justify choice of three used in this study. Also, based on the study results and the fact that all analyzed wake models have some shortcomings, it would be nice to have some idea/suggestion if some other wake model could be more appropriate for this type of events.**

The choice of models was primarily dictated by the page limit. Larsen model did not make it to the list since its performance was similar to the BPA model, except that Larsen model can resolve all cross-sections.

It could be some error in my implementation, but I got wide and weak wake-deficit distribution with Ishihara model for pre-OCC flow, which was not supposed to happen with the model in its original definition. Hence, Ishihara model was not used. I was not satisfied with the performance of Double Gaussian models; the super-Gaussian model does better job on approximating the full wake.

4. **Line 45 and 47: double dots for $x/D = 2..10$ and $r/D = -2..2$ need to be fixed.**

   The formulation is corrected to $x/D \in [2, 10]$ and $r/D \in [-2, 2]$ which should be easier to read.
* * *
**Response to the reviewer comments 2 (RC2)**

**Major revisions**

1. **"We use the results of a multiscale wind-wake modeling during an OCC event at the Alpha Ventus wind farm in the Southern North Sea to study how Gaussian models capture wake deficit variabilities."**

   **Could you also add more clarity in the abstract to where these results come from? Are these results you generated or that have already been published but that you're re-analyzing?**

   Yes, this is one of the simulations produced by our group.

   Re-phrased into:

   *We performed a multiscale wind-wake simulation during an OCC event at the Alpha Ventus wind farm in the Southern North Sea and use its output of the hub height cross-section to study how Gaussian models capture wake deficit variabilities.*

2. **"On the contrary, the Bastankhah and Porté-Agel model and the super-Gaussian model... perform best with different deficit distribution shapes."**

   **Could you please finish this comparison by saying what the shapes should be different from. As written it's unclear if the shapes should be different from those used in the previously mentioned Jensen-Gaussian model, different from those used in previous works, or something else.**

   The sentence was supposed to read that the super-Gaussian model can approximate double peaks in the near wake deficit while still interpreting well the far wake. At the same time, the BPA model is more suitable for single distribution peaks in the far wake.

   Re-phrased into:

   *On the contrary, the Bastankhah and Porté-Agel model and the super-Gaussian model are consistent without tuning. However, the Bastankhah and Porté-Agel model performs better with single-peak velocity deficit distribution in the far wake, while the super-Gaussian model*

*can approximate both double-peak distribution encountered in near wakes and single-peak distribution in far wakes.*

3. **It would be helpful to cite and describe what work has already been done related to this paper and then to clarify how this paper differs from previous work. Right now only one paper is cited in the introduction and it's written by the authors of this paper. In particular, it would be good to discuss any existing literature for (1) tuning wake models, (2) using wake models during rapid transient events esp. OCC, and (3) tuning wake models during rapid transient events esp. OCC. E.g. (2), it would be useful to know whether previous studies have identified which wake models perform well/poorly during rapid transient events. Make sure to specify whether BPA, Jensen-Gaussian, and/or super Gaussian models been tested on rapid transient events? If so, how did they perform? If not, say that to your knowledge none of these models have been tested on rapid transient events. For (3) it would be useful to know whether any previous work has tuned wake models during rapid transient events and, if so, what they found. Furthermore, it would be helpful to know if any wake models have been applied to/tuned for OCC events**

   Yes, to my knowledge, numerical simulation of transient events are rather common, but no study of analytical models in a transient event was performed. For example, Vollmer et al. (2017) used Gaussian function to fit the wake velocity distribution but only for the sake of comparing the simulation result to the lidar data.

4. **This work also frequently references the authors' previous works (Bakhoday-Paskyabi et al., 2022a, b). Could you please explain in the introduction what research those previous works presented and how this paper differs from those works?**

   Those studies describe only the multi-scale simulations. The results from one of it were used here for the model fitting.

5. **Are the high-fidelity numerical simulations results used the result of your previous work (Bakhoday-Paskyabi et al., 2022b) or did you run new simulations not presented in that previous work? Please clarify.**

   **If you ran new simulations, please add sufficient information for a reader to reproduce your work. E.g. you say the LES consists of two nested domains but only specify the dimensions and grid sizes for the inner domain not the outer domain.**

   **If you didn't run new simulations, please double-check that the information you do provide in this section matches that in your previous work. E.g. you say the grid size is $10 \times 10 \times 5\,\mathrm{m}$ but in that previous work you say the grid size is $11 \times 11 \times 5\,\mathrm{m}$.**

   There is an oversight on my part, as the referenced article uses an updated simulation of the same event. The original presentation for DeepWind'22 (not available online) showed the results of the WRF-LES which were later used for the model fitting in this preprint. The WRF-LES updated for the publication was run without the output required to calculate the turbulence intensity, hence, it was unsuitable for the model fitting.

The technical data on the simulation parameters was cut due to the size limit. Now that we are expanding the article, there is sure no limit on adding all of the information about the correct simulation set up.

6. **"The fit uses already known simulation data for the passing period": what do you mean by the passing period? I don't think "passing" is the right word.**

   **"corrected fit": did you try fitting the coefficients to the OCC phase? If so, can you specify how you did that? Based on the results it seems like you tried fitting the coefficients for this phase and it didn't work well; it would be useful to know what you tried.**

   Please, see the next comment.

   **"We also tune coefficients of each model to find the best fit to WRF-LES results" and "The best fit is optimized for all cross-sections in a 10-minute period to avoid tuning models to a specific part of the wake." Do these two sentences use "best fit" to refer only to the "best fit" method or are they referring in general to the idea of fitting model coefficients? If it's the former, please make this more clear. If it's the later, please say something other than "best fit" in those sentences.**

   I had the 'best fit' method in mind when writing 'The best fit is optimized for all cross-sections in a 10-minute period to avoid tuning models to a specific part of the wake'. This part also applies to the 'corrected fit' – it is essentially the 'best fit' of the first 10-minute period of each phase.

7. **"Overall, this similarity allows using AV1 inflow characteristics and thrust coefficient to estimate wakes for both turbines. An ensemble wake is calculated by summing up the normalized wake deficit from the AV1 wake at the regarded cross-section and the deficit at the respective cross-section of the AV4 wake." These two sentences seem to contradict each other. Do you tune the coefficients using the wake estimated from the AV1 as described in the first sentence or using the ensemble wake described in the second sentence?**

   The coefficients are tuned by fitting to the ensemble wake. The inflow characteristics from AV4 are considered to be the same as for AV1. As seen from Fig. 1, the characteristics are generally similar, except for the timestamp 1:40. Using individual characteristics to calculate AV1 and AV4 wakes did not show a strong improvement for model fitting, therefore, we proceeded with a simpler approach by assuming the same flow for both turbines.

8. **Please add the actual equations you use to tune coefficients through the "corrected fit" and "best fit" methods and then describe them. As is, it's unclear how exactly you tuned the model coefficients.**

   Considering the previous comments, Section 3.1 on model fitting will be expanded to clarify the process.

9. **Figure 2: Could you please describe the figure more in the caption? What does each panel refer to? What about the axis? What about the legend? What about the peak transition line?**

[Figure]

Figure 1: 10-minute averaged inflow characteristics at the hub height 2D upstream of AV1 and AV4 (a-c) and the thrust coefficient (d). [Figure 2 of the original preprint]

This comment is not quite clear to me, because most of the information is already contained in the figure. The 'peak transition' may need a clarification – that is the time stamp, where the passing OCC cell affects AV1 and AV4 wakes most. During other periods, the wake is unaffected up to the regarded length of $10D$. Commenting each panel would be duplicating the information from the panel titles, but I will see what can be done when finalizing the revisions.

10. **Figure 2: could you please describe and analyze this figure more fully in the main body of the text? It's only referenced in passing in a single sentence: "The inflow probes for AV4 generally return similar values, except for the time stamp 1:40 (Fig. 2)..." This sentence would be more clear if you took the time to introduce and describe the figure e.g. "Figure 2 shows the 10-minute averaged inflow characteristics upstream of both AV1 and AV4 as well as the thrust coefficient of AV1 and AV4. The figure shows the inflow probes for AV1 and AV4 generally return similar values except for at the time stamp 1:40..."**

The discussion gave me several ideas on which values should be pointed out, so I will definitely expand the description.

11. **Could you provide the results of the model fitting that you described in section 3.1? Aka what values did you get for each coefficient in Table 1 when using the corrected fit and best fit methods? Could extend Table 1 or add additional tables.**

| Model | | | Coefficients | | | | | | | | |
|---|---|---|---|---|---|---|---|---|---|---|---|
| | | | pre-OCC best fits | | | | | OCC best fits | | | |
| | | | 1:10 | 1:20 | 1:30 | 1:40 | 1:50 | 2:00 | 2:10 | 2:20 | 2:30 |
| | | original | (corr.) | | | | | (corr.) | | | |
| BPA | $k_1^*$ | 0.003678 | 0.00371 | 0.00366 | 0.00397 | 0.0038 | 0 | 0.00422 | 0.00394 | 0.0041 | 0.00426 |
| | $k_2^*$ | 0.3805 | 0.3789 | 0.3595 | 0.2193 | 0.3192 | 0 | 0.0672 | 0.2248 | 0.1357 | 0.1182 |
| Jensen-Gaussian | $k$ | 0.05 | 0.011 | 0.019 | 0.029 | 0.02 | -0.01 | 0.002 | 0.012 | 0.008 | 0.004 |
| super-Gaussian | $a_s$ | 0.17 | 0.296 | 0.185 | 0.180 | 0.172 | 0 | 0.216 | 0.465 | 0.319 | 0.3 |
| | $b_s$ | 0.005 | 0.008 | 0.005 | 0.005 | 0.005 | 0.006 | 0.006 | 0.006 | 0.006 | 0.011 |
| | $c_s$ | 0.2 | 0.11 | 0.18 | 0.19 | 0.17 | 0.1 | 0.05 | 0 | 0.002 | 0.02 |

Table 1: Gaussian wake models and their coefficients. The 'original' column corresponds to the coefficients provided in the original definition. The columns under time stamps reflect best fit for each 10-minute period with the first fit of each phase being used for the 'corrected fit' comparison. **(table not final)**

This should be possible (see drafted Table 1), although it may need a font size adjustment or a landscape orientation to fit all the values.

12. **In general, this section would be easier to follow if you take the time to introduce each figure rather than just stating your results and referencing the figure in passing. E.g. rather than "The general behavior of all models follows similar trends (Fig. 3)." say something along the lines of "We calculate the RMSE of. . . and plot these results in Figure 3. The figure shows that the general behavior of all models follows similar trends."**

This is partially caused by the text cuts to fit the page limit and partially by the different approaches to writing. I will keep this comment in mind while expanding and revising the whole article.

13. **"During the peak transition phase at 1:50, . . . Model fitting for the period ending on 1:50 returns unrealistic coefficients, e.g., a negative wake decay coefficient k for the Jensen-Gaussian model."**

   - **Please specify in the text here that for this reason you don't show results in figures 3 and 4 for time 1:50 using the corrected fit and best fit coefficients.**

   - **I would suggest you include the results for time 1:50 using the default coefficients in figures 3 and 4. It would still be interesting to see what the RMSE and what the wake deficit is during the transition and how that varies by wake model.**

The main challenge with 1:50 data is defining the free-flow wind speed to fit the models. As was mentioned in the same paragraph earlier:

*During the peak transition phase at 1:50, the free-flow wind speed becomes ambiguous. The near wake may already be affected by the convective cell – probing the flow upstream would return the correct free-flow wind speed for the near wake. At the same time, the effects of the accelerated flow had not yet reached the far wake – the upstream flow characteristics are not relevant.*

The unified routine of extracting and normalizing the wake deficit across the flow fails here. I was not satisfied with the results when probing the wind speed along the wake instead of the upstream. However, it is worth trying it again and consider 1:50 as a stand-alone case for the sake of illustration and comparison (unified free-flow wind speed upstream vs. new free-flow wind speed for each cross-section).

14. **Figure 3: please describe how the figure is laid out in the caption to make it easier for the reader to orient themselves. Please specify:**

   - **The top three panels use the BPA method; the middle three Jensen-Gaussian, and the bottom super-Gaussian.**

   - **The left three panels show results for pre-OCC and the right three panels results for post-OCC.**

   - **Each column refers to a downstream location.**

   - **Rows are grouped into sets of three corresponding to the RMSE for a time period, e.g. 1:10.**

- **Within each group, the rows are labeled "default", "corr. fit", and "best fit", which correspond to results using the coefficients found using the default, corrected fit, and best fit methods.**

As suggested by RC3, the figure may need a redesign and likely a different approach to present results. If I decide to leave this figure

15. **Figure 3: why do you use different color scales for pre-OCC and post-OCC? It makes it hard to compare the RMSE between the two panels. Could you either provide a compelling reason to use different colors or change them to use the same color?**

    **You say "The agreement to WRF-LES is good for the stabilized flow in the pre-OCC phase but declines as the convective cell approaches the wind farm." That isn't clear to me based on Figure 3. For the pre-OCC panels, the RMSE goes up to 2.0. In contrast, for the post-OCC panels, the RMSE only goes up to .15. Doesn't this mean that the error is actually higher for the post-OCC panels than the pre-OCC panels?**

    The RMSEs shown are the absolute values of the RMSE for normalized wake deficit distribution. The normalized wake deficit for pre-OCC is generally larger (up to 0.75 vs. up to 0.2 for post-OCC). Hence, the absolute error is also larger for pre-OCC cases, especially for cross-sections with double peaks where the models have the strongest discrepancy.

    I was considering a normalization of RMSE, but could not choose a normalization value that would produce comparable results. The BPA model could be a good choice if it returned a distribution for every cross-section (in the current formulation used, it fails for high $C_T$ at around $x/D < 2.5$). Besides, normalizing the RMSE of the normalized wake deficit could overcomplicate the explanation. Still, it is a good point to consider for making the comparison easier to understand from the figure.

16. **Figure 4: similar to for figure 3, could you please describe the figure more in the caption to help orient the reader. Please talk about what each column and row of figures is and then within each figure please define what the horizontal and vertical axes are. Please also describe the legend aka what does each line refer to.**

    The description will be expanded.

17. **Figure 4: can you please discuss the results from all the columns? 4b and 4d are never discussed.**

    The wake models behave much more similar after 2:00 compared to the pre-OCC wake distributions. Therefore, I think, referencing 4b, 4d, and 4f at once when starting to discuss the OCC's effect on the wakes would solve the problem.

18. **Could you please discuss how you envisioned the corrected fit extending to simulations with more turbines and for longer time periods? Do you imagine tuning the coefficients for each turbine once before and once after each OCC event?**

    The coefficient re-fitting was used to check whether the model can potentially describe the wake better. This approach proved to be necessary for the Jensen-Gaussian model, while

the other two of the regarded models performed reasonably in their suggested formulation. Considering the results, I do not think that the Jensen-Gaussian model would be a good choice to apply in a transient event. However, I see that this study needs a discussion on whether it is worth adjusting the coefficients of the BPA and super-Gaussian models for the OCC event or we can be satisfied by their estimation as it is.

19. **How will you identify when an OCC event is about to occur? How will you identify when an OCC event has just finished?**

    The identification of an OCC event goes beyond the narrow topic of the article, as it is more related to the short-term forecast and weather monitoring. However, the OCC cells do not occur as random as gusts and can be observed for awhile. If we are aware of an OCC cell moving towards the wind farm, then a substantial change of the characteristics (as observed in Figure 2) would be a signal for the OCC event starting at the wind farm.

**Spelling and grammar**

1. **Abstract: "We use the results of a multiscale wind-wake modeling during an OCC event at the Alpha Ventus wind farm in the Southern North Sea to study how Gaussian models capture wake deficit variabilities." This sentence should be "the results of multiscale wind-wake modeling" (no a) or "the results of a multiscale wind-wake model".**

    As noted above, this sentence was changed into

    *We performed a multiscale wind-wake simulation during an OCC event at the Alpha Ventus wind farm in the Southern North Sea and use its output of the hub height cross-section to study how Gaussian models capture wake deficit variabilities.*

    in order to reflect that we are analyzing results from our own simulation.

2. **First paragraph of the introduction: either refer to engineering wake models as plural or singular throughout for consistency aka either say "engineering wake models... models'... the models... models' " or say "an engineering wake model... model's... the model... model's ".**

    **Introduction paragraph 3: it should be "the models' coefficients are corrected" since you're referring to multiple models.**

    Corrected to plural for all instances.

3. **Figures 1 and 2: Typically units are reported in parenthesis or brackets; could you switch to this convention? E.g. rather than Y, m say "Y [m]".**

    Corrected

4. **Figure 1: Could you please use complete sentences in your caption? E.g. "the OCC event is about to begin, the wind farm is not yet affected" is not a complete sentence; it's a comma splice.**

    Corrected to

    *a) Flow before the OCC event. b) The OCC event is about to begin, the wind farm is not yet affected. c) The flow in the wind farm undergoes radical changes due to the OCC*

*event. d) The convective cell had engulfed the whole wind farm, the flow is temporarily stabilizing.*

5. **Figure 2: Could you change the label to time stamp rather than time for consistency with the rest of your discussion?**

   Corrected for the axis label. The full time stamp is too long to be used as ticks labels.

6. **Methodology 3.2: "is defined similarly to the Jensen model" should be "is defined similarly to in the Jensen model"**

   I couldn't find any examples of using 'similarly to in', there might be some typo in the comment. The part

   *The only coefficient to fit, the wake decay coefficient k, is defined similarly to the Jensen model...*

   is re-phrased into

   *The only coefficient to fit, the wake decay coefficient k, is defined the same as in the Jensen model...*

7. **Methodology 3.2: "less parameterized coefficients." should be "fewer parameterized coefficients."**

   Corrected

8. **Methodology 3.2: "The label 'T' ". Fix the first apostrophe so it curls the correct direction.**

   Corrected. However, this change may become obsolete if this figure is replaced completely.

9. **Methodology 3.2: "proposes two method" should be "proposes two methods" (methods b/c it's plural.)**

   Corrected

10. **You're often missing articles before nouns. E.g. "at 1:50 time stamp" should be "at the 1:50 time stamp". Could you please check throughout your article for any nouns missing articles and add the appropriate article, e.g. "a", "an", or "the"? Commas are only used before "but" when they separate two independent clauses not when connecting an independent clause and a sentence fragment. E.g. "low thrust, but high wind speed" should not have a comma. Please check all uses of ", but" and correct where there shouldn't be a comma.**

    **Please check for typos. E.g. "The turbulence intensity fluctuates withing", "withing" should be "within". "x/D = 2..10"and "r/D = -2..2" -¿ not sure what this should be but something is off. "We did not account the deflection" should be "We did not account for the deflection" (you need to say "for"). "maximums positions" should be "maximum positions".**

    Thank you for pointing this out, I will keep it in mind while revising the article.

    "x/D = 2..10"and "r/D = -2..2" were changed into $x/D \in [2, 10]$ and $r/D \in [-2, 2]$ to avoid confusion.

**Response to the reviewer comments 3 (RC3)**

**Major revisions**

1. **The wakes predicted by WRF-LES and by Gaussian models are compared for only a single OCC. How sure can the authors be that the results in the manuscript are a good representation of the disagreement between models? I would hesitate to draw any real conclusions about the possible benefits of model parameter tuning or the underlying cause for differences in wake model predictions from a single sample.**

   We discussed whether we can add another transient case – the LLJ simulation. Although, it is not as interesting in terms of wake interaction. I am leaning towards regarding other turbines in the OCC simulation and also regard the whole farm.

   I tried the same models on the lidar data for a different period with no transient event to check that the super-Gaussian model had not accidentally matched PALM's equations for wake simulation. The results were in line with what I saw for the pre-OCC phase, but I am hesitating to include them into the paper as it may diverge the focus from a transient event. Sadly, we do not have lidar data for the OCC event, so it is not possible to run the same procedure on the observations and compare the results to the simulation.

2. **Similarly, the study looks at a small region of the flow in Alpha Ventus, containing two wakes that interact strongly. The study design does not take into account that there are several possible superposition methods that could be used for analytical wake models, or the possibility of added prediction uncertainty introduced by superposition (see ref 1, below) .**

   There are a couple of drawbacks that should be unaffected by the superposition method. First, the cross-sections of the AV1 wake are little affected by the nearby turbines; basically, we are dealing with a single wake for $x/D \leq 6$. Second, the BPA model in the formulation used does not resolve near wakes ($x/D \leq 2.5$) for high $C_T$ and $I_a$ in the pre-OCC phase – it does not return any distribution for near-rotor cross-sections of the AV4 wake to combine it with the AV1 wake. (Side note: it may be worth to consider rolling back to $\varepsilon = 0.25\beta$ instead of $\varepsilon = 0.2\beta$ which causes the failure; although, $\varepsilon = 0.2\beta$ definition was considered to be more accurate by the original authors). Hence, I am unsure whether the model behavior would gradually change with a different superposition method. However, if we expand the article in the direction of reviewing wake from other turbines inside the wind farm, then considering different superposition methods for the comparison would be a logical move.

   **Other engineering models exist that do not rely on a superposition scheme. Have the authors considered using those to limit uncertainty to only the definition of the wake itself? (see refs 2–4).**

   From what I see, using the curled wake model would require a different pipeline from what I used for this study. We were aiming for a small article originally, so our choice of models was primarily driven by the code and data produced in the previous works. It would be interesting to try the curled wake model in a follow-up article as the OCC case definitely has more to offer for the study.

3. **Figure 1. The cross-section of the inner**

What do the small makers downstream of AV1 indicate? Sometimes they over-lap AV4 and other times they do not. Do these point out measurement areas? This does not match the 17 cross sections from the simulations described in the text.

These are the dotted lines representing the cross-sections. Plotting all 17 cross-sections would clutter the plot, so only every 4th cross-section was plotted to illustrate the extent of the extracted wake deficit distributions. This definitely needs a clarification, thank you.

4. **Figure 2. 10-minute averaged inflow chara I would be interesting to include AV2 in the subplots of Figure 2. AV2 is already called out in Figure 1 and comparing it to AV1 may highlight the spatial variability in turbine performance as well as inflow characteristics.**

Apparently, there is little difference between AV1 and AV2 10-minute averaged inflow (Fig. 2). In general, the turbines show the strongest discrepancy for 1:40 (most turbines are affected by the wakes) and 1:50 (some turbines are already affected by the OCC cell, some are not). Other deviations are weaker and of the order of ones observed for 2:30.

5. **Is PALM a normal LES framework to use within the WRF model? It would benefit the readers to explain a bit more of the details in the simulation scheme. For example, how is turbulence information exchanged between the WRF simulation and PALM? Do you use the cell-perturbation method? Have you tested sensitivity? Is the WRF simulation driven with reanalysis data? If so, what is the source?**

Until recently, PALM could be only coupled with the COSMO-DE model (Kadasch et al., 2021). Coupling with WRF is a recent addition described in Lin et al. (2021), it was already used for an urban boundary layer simulation (Vogel et al., 2022). The turbulence is added via PALM's own synthetic turbulence generator. While I was writing this response, an article from our research group was published covering WRF coupling with PALM in-depth (Ning et al., 2023). They have observed a slight bias in the simulated TKE, which can be attributed to PALM's turbulence generator. And, of course, the smallest turbulence scales would not be resolved by the LES due to grid-spacing restrictions related.

I was not considering (Vollmer et al., 2017) as a possible reference, since they were using COSMO-DE input data instead of coupling with WRF. Nevertheless, it would be worth mentioning as another example of running PALM LES with a dynamic input from another NWP model.

Sadly, we do not have lidar scans of FINO1 for the period of this OCC event, so it is not possible to get data on the actual wakes.

6. **Figure 3. This is an interesting figure and appears to be the main result of the study. However, trying to compare results from the various models and model tunings based on the colors of each cell in the figures makes the actual data feel somewhat arbitrary and qualitative. Perhaps the figure could be replaced with trend lines comparing either the different models or the different tuning strategies. I think this would also provide a better opportunity to discuss the surprising result of the peaks in RMSE for the different models.**

[Figure]

Figure 2: 10-minute averaged inflow characteristics at the hub height $2D$ upstream of AV1, AV2, and AV4 (a-c) and the thrust coefficient (d).

**Minor comments**

1. **per-Gaussian model proposes two method of finding n(x): root-solving a**
   **should be plural, "methods"**

   Corrected

2. **\*= 0.2$\sqrt{\beta}$, $\beta$ = 121 + $\sqrt{1}$ -CT$\sqrt{1}$ -CT (4)**
   **Combine fractions to make equation (4) more readable.**

   The equation is corrected to

   $$\varepsilon = 0.2\sqrt{\beta}, \ \beta = \frac{1 + \sqrt{1 - C_T}}{2\sqrt{1 - C_T}} \tag{1}$$

3. **as= 0.17, bs= 0.005, cs= 0.2 by the direct hit from AV1 wake . Overall, this**
   **similarity allow**

   **Rephrase.**

   The sentence is rephrased to a simpler statement:

   *The inflow probes for AV4 generally return similar values, except for the time stamp 1:40 (Fig. 2) – the wind speed and turbulence intensity in front of AV4 are strongly affected by the AV1 wake.*

**References**

E. Kadasch, M. Sühring, T. Gronemeier, and S. Raasch. Mesoscale nesting interface of the palm model system 6.0. *Geoscientific Model Development*, 14:5435–5465, 9 2021. ISSN 1991-9603. doi: 10.5194/gmd-14-5435-2021.

D. Lin, B. Khan, M. Katurji, L. Bird, R. Faria, and L. E. Revell. Wrf4palm v1.0: a mesoscale dynamical driver for the microscale palm model system 6.0. *Geoscientific Model Development*, 14:2503–2524, 5 2021. ISSN 1991-9603. doi: 10.5194/gmd-14-2503-2021.

X. Ning, M. B. Paskyabi, H. H. Bui, and M. M. Penchah. Evaluation of sea surface roughness parameterization in meso-to-micro scale simulation of the offshore wind field. *Journal of Wind Engineering and Industrial Aerodynamics*, 242:105592, 11 2023. ISSN 01676105. doi: 10.1016/j.jweia.2023.105592.

J. Vogel, A. Afshari, G. Chockalingam, and S. Stadler. Evaluation of a novel wrf/palm-4u coupling scheme incorporating a roughness-corrected surface layer representation. *Urban Climate*, 46:101311, 2022. ISSN 2212-0955. doi: https://doi.org/10.1016/j.uclim.2022.101311.

L. Vollmer, G. Steinfeld, and M. Kühn. Transient les of an offshore wind turbine. *Wind Energy Science*, 2:603–614, 12 2017. ISSN 2366-7451. doi: 10.5194/wes-2-603-2017.